# KnowProxy: Adapting Large Language Models by Knowledge-guided Proxy

**Gukhyeon Lee**[1*]   **Yeachan Kim**[2*]   **SangKeun Lee**[1,3]
[1]Department of Artificial Intelligence, Korea University, Seoul, Republic of Korea
[2]Division of Language & AI, Hankuk University of Foreign Studies, Seoul, Republic of Korea
[3]Department of Computer Science and Engineering, Korea University, Seoul, Republic of Korea
`gukhyeon_lee@korea.ac.kr`  `yeachan@hufs.ac.kr`  `yalphy@korea.ac.kr`

## Abstract

Adapting large language models (LLMs) using smaller proxy models has been shown to improve training efficiency, where the LLMs remain frozen while the proxies are tuned on top. However, this approach typically requires access to the output probability distributions of LLMs, which are often inaccessible or unstable. To address this limitation, we propose KnowProxy, a knowledge-guided proxy framework in which the proxy is trained with textual knowledge rather than probability distributions. Specifically, we first elicit textual knowledge and reasoning from frozen LLMs through prompting, and then the proxy model learns to adapt this reasoning to target task distributions. We evaluate KnowProxy on diverse reasoning benchmarks with different fine-tuning scenarios. Comprehensive results show that KnowProxy achieves competitive or even better performance without direct access to probability distributions, thereby providing a scalable and versatile alternative to traditional fine-tuning.[1]

## 1 Introduction

Large Language Models (LLMs) have achieved remarkable performance across a variety of NLP tasks (OpenAI et al., 2024; Team et al., 2025; Grattafiori et al., 2024). To ensure that such LLM behaviors align with human intentions and the specific requirements of downstream tasks, fine-tuning plays a crucial role. However, their computational demands make direct fine-tuning of these LLMs resource-intensive and often impractical for many cases (Zhao et al., 2024; Miles et al., 2024). Moreover, when dealing with proprietary LLMs, which are frequently closed-source and inaccessible, direct modification becomes impossible.

One promising approach to addressing this limitation is to train smaller proxy language models on top of the outputs from LLMs (Liu et al., 2024; Ormazabal et al., 2023). For example, proxy-tuning (Liu et al., 2024) adjusts LLM's outputs via a lightweight proxy that reweights the LLM's probability distributions. Similarly, CombLM (Ormazabal et al., 2023) trains a smaller model separately and combines its predictive distribution with that of the LLM. Despite their effectiveness, these approaches share two critical limitations: (i) They assume access to the full probability distributions and a shared vocabulary between LLMs and proxies, restricting their applicability to black-box LLMs that provide only textual outputs. (ii) Recent studies show that LLM-generated probability distributions are often unstable and unreliable (Atil et al., 2024; Gu et al., 2024), which can degrade downstream performance.

To overcome these limitations, we propose KnowProxy, a knowledge-guided proxy framework for adapting LLMs without relying on their probability distributions. Instead of accessing predictive probabilities, KnowProxy elicits textual knowledge and reasoning from frozen LLMs through prompting. A lightweight proxy model is then optimized on the elicited knowledge together with the input query, learning to map LLM-derived reasoning and knowledge into the target task distribution. This design enables adaptation even for proprietary black-box LLMs, while also mitigating instability by avoiding reliance on probability distributions.

---

[*]Equal contribution.
[1]Our code and data are available at `https://github.com/2gukhyeon/KnowProxy.git`.

A remaining challenge for proxy-based adaptation is the additional inference cost introduced by always involving the proxy model. To address this, KNOWPROXY incorporates a dynamic routing mechanism that adaptively determines when the proxy is required. Specifically, we elicit uncertainty scores for the LLM's generated reasoning and knowledge, and use these scores to decide whether to invoke the proxy model. In this way, KNOWPROXY maintains efficiency by selectively engaging the lightweight model only when the LLM's outputs are deemed unreliable.

We evaluate KNOWPROXY on a diverse set of reasoning benchmarks under multiple configurations. Experimental results show that KNOWPROXY consistently outperforms existing proxy-based methods, achieving superior accuracy and robustness while requiring no direct access to probability distributions. Notably, it delivers strong performance even in black-box settings where conventional fine-tuning is infeasible. In summary, the contributions of this work include the following:

○ We introduce KNOWPROXY, a novel proxy-based fine-tuning framework that adapts LLMs through textual knowledge and reasoning rather than probability distributions, enabling applicability to black-box settings.
○ We integrate a dynamic routing mechanism into KNOWPROXY, which adaptively activates the proxy model only when necessary, thereby enhancing efficiency while preserving accuracy.
○ We demonstrate that KNOWPROXY outperforms proxy-based methods and achieves performance comparable to direct fine-tuning, highlighting its practical value in fine-tuning scenarios.

## 2 RELATED WORKS

### 2.1 FINE-TUNING LARGE LANGUAGE MODELS BY PROXY

As billion-scale LLMs increasingly dominate applications and research communities, fine-tuning them has become even more challenging—even with parameter-efficient methods such as low-rank adaptation (LoRA) (Hu et al., 2022) and adapters (Houlsby et al., 2019). To address these challenges, an alternative line of work has investigated proxy-based approaches, where smaller language models (i.e., proxy) are trained on the outputs of frozen LLMs to adapt them to the target domain (Liu et al., 2024; Ormazabal et al., 2023). For instance, CombLM (Ormazabal et al., 2023) trains a separate smaller model and combines its predictive distribution with that of the LLM. Similarly, proxy-tuning (Liu et al., 2024) employs a lightweight model to reweight the predictive distributions of LLMs. However, these methods require access to the probability distributions of LLMs and assume a shared vocabulary space between the LLMs and the smaller model.

Compared to these methods, KNOWPROXY has distinct properties. Instead of relying on the probability distributions of LLMs, KNOWPROXY leverages textual knowledge and reasoning elicited from frozen LLMs. This design makes it directly applicable to black-box settings, where only text outputs are available. Moreover, by shifting from probability distributions to textual representations, KNOWPROXY avoids the instability and unreliability issues often observed in LLM-generated distributions (Atil et al., 2024; Gu et al., 2024). The proxy model is thus trained to internalize and adapt reasoning expressed in text, yielding more stable and transferable performance across tasks. Finally, KNOWPROXY integrates an adaptive routing mechanism, ensuring that the proxy is invoked only when additional reasoning is required, thereby improving both efficiency and robustness.

### 2.2 ELICITING UNCERTAINTY FROM LANGUAGE MODELS

Assessing the confidence (or uncertainty) elicited from LLMs is crucial for improving the factuality of their responses and enhancing the quality of generated text (Geng et al., 2024). Previous studies have primarily extracted uncertainty from the output probabilities (Mielke et al., 2022; Kuhn et al., 2023; Duan et al., 2024). However, this approach faces applicability constraints, as it cannot be applied to black-box models whose internal probabilities are inaccessible. To address this limitation, several studies have proposed prompting-based methods to compute uncertainty scores directly from LLM textual outputs—applicable to both open-source and API-based models—showing that these scores correlate well with model performance and output quality (Tian et al., 2023; Xiong et al., 2024; Dong et al., 2024).

Building on these studies, we use elicited uncertainty scores primarily for adaptive routing. Unlike prior proxy-based methods that always invoke the proxy model, KNOWPROXY selectively engages

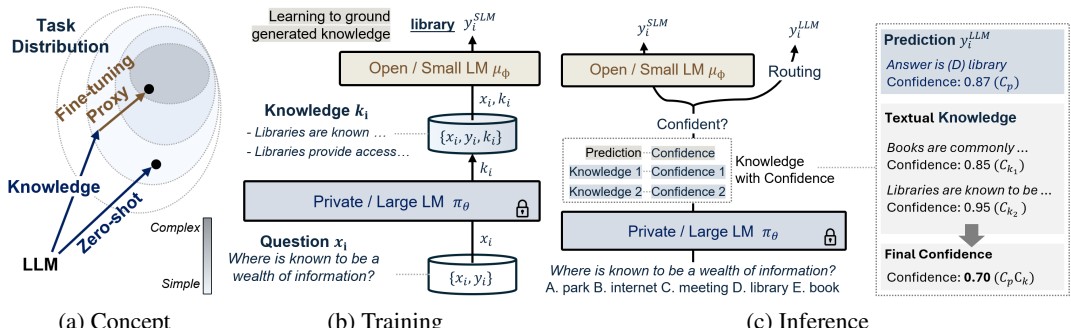

Figure 1: Conceptual illustration of KNOWPROXY. (a) A smaller proxy model is trained using textual knowledge generated by LLMs to capture the target task distribution better. (b) During training, knowledge is elicited from LLMs via prompting and incorporated as an auxiliary input to the proxy. (c) At inference, KNOWPROXY utilizes multiple confidence scores from LLMs to perform dynamic routing, thereby balancing efficiency and accuracy.

it only when the LLM's outputs are judged uncertain or unreliable, thereby reducing inference overhead while maintaining robustness. Moreover, whereas previous work has considered uncertainty only at the prediction level, our design estimates uncertainty for each piece of elicited knowledge, allowing finer-grained routing decisions and more stable adaptation.

## 3 KNOWPROXY: KNOWLEDGE-GUIDED PROXY

We introduce KNOWPROXY, a framework that adapts LLMs using smaller proxy models based on textual knowledge and reasoning elicited from frozen LLMs. We first outline the problem setup for LLM adaptation (§3.1), then describe how we elicit textual knowledge and reasoning from LLMs and train proxy models to leverage the extracted textual representations while aligning with the target task distribution (§3.2). Finally, we present an adaptive routing mechanism that reduces inference overhead by selectively invoking the proxy model only when needed (§3.3). Figure 1 illustrates the workflow of KNOWPROXY.

### 3.1 PRELIMINARIES AND PROBLEM FORMULATION

We first revisit the traditional fine-tuning and define the problem setup in our method.

**Direct Fine-tuning.** A straightforward way to train LLMs is to perform direct fine-tuning using a supervised objective defined over a training dataset $\mathcal{D} = \{(x^i, y^i)\}_{i=0}^{N-1}$, where $N$ denotes the dataset size. The fine-tuning objective is formulated as:

$$\min_{\theta} -\mathbb{E}_{(x,y)\sim\mathcal{D}} \left[\log \pi_\theta(y \mid x)\right] \tag{1}$$

Here, $\pi_\theta$ denotes the language modeling function of the LLM parameterized by $\theta$. However, directly fine-tuning $\theta$ is often impractical due to two major challenges: (1) the massive scale of $\theta$, which renders optimization computationally prohibitive, and (2) the restricted accessibility of $\theta$ in black-box LLMs, where the underlying parameters are not exposed.

**Proxy-based Fine-tuning.** To overcome these challenges, proxy-based fine-tuning approaches have been proposed, in which a smaller, accessible model $\mu_\phi$, parameterized by $\phi$, serves as a proxy for the fine-tuned LLM (Ormazabal et al., 2023; Liu et al., 2024). The training objective is reformulated as:

$$\min_{\phi} -\mathbb{E}_{(x,y)\sim\mathcal{D}} \left[\log \mu_\phi(y \mid x)\pi_\theta(y \mid x)\right] \tag{2}$$

By optimizing the smaller model $\mu_\phi$ on the dataset using the LLM's predictive distributions, this approach effectively performs fine-tuning through the proxy without requiring access to the LLM's parameters.

However, this approach has two key limitations rooted in its reliance on probability distributions: (i) it requires access to the predictive distributions of LLMs, which are often unavailable, and (ii) even when accessible, these distributions are frequently unstable (Atil et al., 2024; Gu et al., 2024). To address this, we reformulate proxy-based fine-tuning to leverage textual knowledge generated by LLMs, making it applicable even in black-box scenarios. Given an input $x$, the LLM generates knowledge $k$ according to $\pi_\theta(k \mid x)$, where $k$ denotes textual knowledge or reasoning extracted from the model. This leads to the following knowledge-guided objective:

$$\min_\phi -\mathbb{E}_{(x,y)\sim\mathcal{D}} \left[\log \mu_\phi(y \mid x, k)\right] \text{ where } k \sim \pi_\theta(k|x)$$

Here, $k$ represents the generated textual knowledge that provides additional context for solving the query $x$. This reformulation enables effective fine-tuning through proxies without relying on the LLM's predictive distributions. Moreover, because the proxy model directly learns to leverage knowledge generated by the LLM, it can internalize this information in a more stable and controlled manner. In the following sections, we describe how textual knowledge is elicited from LLMs and how the proxy model is trained to align this knowledge and reasoning to target tasks.

## 3.2 Adaptation with Knowledge-guided Proxy

**Knowledge and Reasoning Generation.** To optimize the reformulated training objective (Eq. equation 9), KNOWPROXY begins by eliciting textual knowledge for the training dataset $\mathcal{D}$. In this work, we define knowledge as the relevant cues required to solve a given problem, such as underlying principles (Cai et al., 2025), reasoning steps (Wei et al., 2022), or relevant facts (Park et al., 2024). This knowledge is obtained through knowledge-eliciting prompts to LLMs. Along with the knowledge, we also extract a confidence score for each piece, which serves as an estimate of its reliability and allows weighting according to its expected correctness. This process can be formally represented as follows:

$$k, c = \pi_\theta(P_k, x) \tag{3}$$

Here, $P_k$ denotes the knowledge-eliciting prompt[2] (see Appendix §B.3 for the complete list of prompts), while $k$ and $c \in [0, 1]$ represent the extracted textual knowledge and its associated confidence score for solving the given query $x$. Importantly, rather than relying on a single output, we generate multiple knowledge–confidence pairs. This design captures the diversity of potential reasoning paths and mitigates the risk of over-reliance on any individual extraction.

However, the knowledge generated by LLMs is not always reliable for solving the given query, as it may contain hallucinations or irrelevant information. To mitigate this issue, we apply a filtering process to the generated knowledge, defined as follows:

$$k = \{k_i \mid (k_i, c_i) \in \mathcal{K}, c_i > \alpha\}, \tag{4}$$

Here, $\mathcal{K}$ denotes the set of knowledge–confidence pairs for the given query, and $\alpha$ is a predefined threshold that specifies the minimum confidence level required to retain knowledge[3]. After applying this filtering process, we construct the knowledge-augmented training dataset $\mathcal{D}_{\mathcal{K}} = \{(x^i, k^i, y^i)\}_{i=0}^{N-1}$, where each training instance $(x^i, k^i, y^i)$ consists of the original input $x^i$, the corresponding filtered knowledge set $k^i$, and the target output $y^i$.

**Proxy Optimization with Generated Knowledge.** Based on the knowledge-augmented dataset, we train a smaller language model to leverage the reasoning and knowledge elicited from LLMs while aligning with the target task distribution, using a single objective. Given a query $x$ and its associated knowledge $k$, we construct an augmented input by concatenating $x$ and $k$, and train the proxy model $\mu_\phi$ with a standard supervised objective. This training procedure enables the proxy to map LLM-derived reasoning into task-specific outputs, thereby adapting the frozen LLM to downstream requirements. Further details on the knowledge adaptation process are provided in Appendix A.

---

[2]Based on our empirical analysis (See Appendix §C.1 for a more detailed analysis), we adopt the decomposition-style prompt design to elicit knowledge from LLMs.

[3]The threshold for knowledge filtering is empirically determined, and the detailed analyses can be found in Appendix §C.2.

### 3.3 ADAPTIVE REASONING IN KNOWPROXY

Although adapting smaller models to fine-tune LLMs through proxies is effective, it incurs additional inference costs for every query. A more efficient strategy is to invoke the proxy model only when necessary—specifically, when the LLM alone is unlikely to yield a correct output. To achieve this, we incorporate a dynamic routing mechanism into the inference phase of KNOWPROXY.

**Confidence Elicitation.** To implement the dynamic routing process, we first obtain the LLM's prediction for the given query along with its associated confidence score $C_{\mathbf{prediction}}$. We efficiently achieve this by augmenting the knowledge-eliciting prompt $P_k$ (Eq. equation 3) with instructions to elicit both the prediction and its confidence. For the routing decision, we incorporate not only the prediction confidence but also the confidence scores of the generated knowledge. This is because the reliability of the generated knowledge provides additional insight into whether the LLM's reasoning is sound; low-confidence knowledge may indicate uncertainty or hallucination, signaling the need to invoke the small model to improve accuracy. We thus derive the final confidence scores as follows:

$$C_{\text{final}} = C_{\text{knowledge}} \cdot C_{\text{prediction}}, \text{ where } C_{\text{knowledge}} = \prod_{k=1}^{K} c_k \tag{5}$$

Here, $K$ denotes the predefined number of generated knowledge instances. Since multiple pieces of knowledge are elicited per query, we aggregate their confidence scores into a single reliability measure[4]. Note that when aggregating the final confidence score, we include all confidence scores—even those from filtered knowledge—to capture the LLM's comprehensive understanding of the query.

**Adaptive Reasoning Paths.** KNOWPROXY determines whether to rely directly on the LLM's prediction or to engage the proxy model trained to leverage generated knowledge. If the aggregated confidence score (Eq. equation 5) exceeds a predefined threshold, the system outputs the LLM's prediction as the final answer without invoking the proxy. Conversely, if the confidence score falls below the threshold, KNOWPROXY activates the proxy model, which incorporates the generated knowledge to refine the prediction. This decision process can be formalized as follows:

$$y = \begin{cases} \pi_\theta(y|x), & \text{if } C_{\text{final}} \geq \tau \\ \mu_\phi(y \mid x, k \sim \pi_\theta(k|x)), & \text{if } C_{\text{final}} < \tau \end{cases} \tag{6}$$

Here, $\tau$ denotes the predefined threshold[5]. Through this adaptive reasoning path, KNOWPROXY dynamically balances computational efficiency and prediction accuracy by invoking the proxy model only when the LLM's prediction is deemed unreliable. This design ensures that inference remains efficient while maintaining accurate predictions across diverse queries.

## 4 EXPERIMENTS

In this section, we evaluate KNOWPROXY to verify its efficacy on fine-tuning scenarios. Specifically, we aim to answer the following research questions:

○ Does KNOWPROXY achieve performance comparable to existing proxy-based methods?
○ Is KNOWPROXY effective across diverse fine-tuning scenarios, including black-box LLMs?
○ Can KNOWPROXY effectively reduce the additional costs introduced by proxy-based training?

### 4.1 EXPERIMENTAL SETUPS

**Datasets.** We evaluate KNOWPROXY on a broad suite of complex reasoning benchmarks, including OpenBookQA (Mihaylov et al., 2018), ARC-Challenge (Clark et al., 2018), CommonsenseQA (Talmor et al., 2019), QASC (Khot et al., 2020), PhysicalIQA (Bisk et al., 2020), SocialIQA (Sap et al., 2019), Winogrande (Sakaguchi et al., 2020), BoolQ (Clark et al., 2019), and StrategyQA (Geva et al., 2021). We also include distinct benchmarks such as TruthfulQA (Lin et al., 2022), ScienceQA (Lu et al., 2022), and mCSQA (Sakai et al., 2024).

---

[4]In Section 4, we demonstrate that our confidence aggregation strategy yields more reliable estimates than existing confidence elicitation methods.

[5]The routing threshold is empirically determined, and the detailed analyses can be found in Appendix §C.2.

| Method | OBQA | ARC$_h$ | PIQA | CSQA | QASC | SIQA | WNGR | StrategyQA | BoolQ | Avg. |
|---|---|---|---|---|---|---|---|---|---|---|
| Fine-tuning LLM | 82.2 | 76.2 | 87.7 | 79.5 | 82.9 | 80.5 | 87.3 | 71.5 | 86.9 | 81.6 |
| Fine-tuning SLM | 73.2 | 60.9 | 80.3 | 72.0 | 68.0 | 74.9 | 75.4 | 66.5 | 85.4 | 73.0 |
| *Advanced Zero-shot Reasoning (Frozen LLM)* | | | | | | | | | | |
| Zero-shot | 72.2 | 68.6 | 75.8 | 67.7 | 75.9 | 65.3 | 53.6 | 60.9 | 78.3 | 68.7 |
| Self-Talk | 74.8 | 74.2 | 75.9 | 72.3 | **80.5** | 67.0 | 54.6 | 57.6 | 78.5 | 70.6 |
| Chain-of-Thought | 77.6 | **80.0** | 75.6 | 73.1 | 79.0 | 68.6 | 57.8 | 69.0 | 76.7 | 73.1 |
| Plan-and-Solve | 76.6 | 75.3 | 74.3 | 73.7 | 79.8 | 66.4 | 58.2 | 66.8 | 73.9 | 71.7 |
| *Proxy-based Training (Frozen LLM + SLM)* | | | | | | | | | | |
| Proxy-tuning | 77.2 | 69.6 | 80.1 | 70.8 | 69.9 | 72.6 | 65.7 | 64.6 | 76.2 | 71.9 |
| CombLM | 78.6 | 72.6 | 81.1 | 72.5 | 76.9 | 73.7 | 69.3 | 67.2 | 76.8 | 74.3 |
| BBox-Adapter | 76.2 | 68.6 | 73.8 | 73.3 | 73.8 | 72.7 | 53.7 | 69.0 | 70.5 | 70.2 |
| KNOWPROXY (ours) | **80.2** | 75.2 | **83.4** | **75.0** | 78.1 | **76.3** | **77.8** | **72.9** | **85.1** | **78.2** |

Table 1: Evaluation results for test accuracy (%) on nine reasoning benchmarks. The best and second-best results are highlighted in **boldface** and underlined, respectively. In these experiments, we use Llama-3.2 (3B) as the frozen LLM and Llama-3.2 (1B) as the smaller proxy (SLM).

**Baselines.** We primarily compare KNOWPROXY with recent proxy-based approaches, including CombLM (Ormazabal et al., 2023) and Proxy-tuning (Liu et al., 2024). We also include BBox-Adapter (Sun et al., 2024), which, while not a proxy-based method, trains a smaller evaluation model to select better answers from multiple samples generated by the LLM. This provides a complementary baseline, as it adapts LLM outputs through answer selection rather than distributional alignment. In addition, we evaluate against established reasoning-based approaches, i.e., Self-talk (Shwartz et al., 2020), Zero-shot-CoT (Kojima et al., 2022), and Plan-and-Solve (Wang et al., 2023).

**Backbone.** To demonstrate the applicability of KNOWPROXY across diverse LLMs, we evaluate it on two categories: API-based models (ChatGPT (GPT-3.5-turbo) and GPT-5 (mini)) and open-source models (i.e., Llama-3.2-3B-Instruct[6], Qwen3-4B-Instruct-2507[7], Mistral-7B-Instruct-v0.2[8], and Llama-2-13B-Chat (Touvron et al., 2023). Furthermore, to showcase the adaptability of KNOWPROXY, we conduct experiments using various small models, such as Llama-3.2-1B-Instruct[9], LaMini-GPT-774M (Wu et al., 2024), Qwen2.5-0.5B-Instruct (Qwen et al., 2025), and Pythia family (Biderman et al., 2023), covering a range of model sizes and model families. The experimental details are provided in Appendix B.

## 4.2 MAIN RESULTS

To validate the effectiveness of our approach, we first compare KNOWPROXY with existing proxy-based methods that adapt LLMs by redistributing their predictive distributions through lightweight models (e.g., Proxy-tuning, CombLM) or by generating adapted responses via multi-step beam search, where candidate responses are ranked by small models (e.g., BBox-Adapter). As shown in Table 1, KNOWPROXY consistently outperforms these methods across all reasoning benchmarks by a substantial margin. Notably, KNOWPROXY even achieves accuracy comparable to direct fine-tuning of LLMs on several tasks (e.g., OpenBookQA, ARC-Challenge, StrategyQA, and BoolQ)—a performance level not previously attained by proxy-based approaches. Additionally, we observe that KNOWPROXY is effective even on domain-specific tasks, including generative tasks. More detailed results can be found in the Appendix C.3.

We further observe that existing proxy-based methods sometimes underperform even the zero-shot reasoning capabilities of LLMs on certain benchmarks (e.g., QASC and BoolQ), likely due to the instability of LLM probability distributions. In contrast, KNOWPROXY delivers substantial improvements on these tasks, indicating that proxies trained on textual representations provide greater robustness and reliability than distribution-based approaches.

---

[6]https://huggingface.co/meta-llama/Llama-3.2-3B-Instruct

[7]https://huggingface.co/Qwen/Qwen3-4B-Instruct-2507

[8]https://huggingface.co/mistralai/Mistral-7B-Instruct-v0.2

[9]https://huggingface.co/meta-llama/Llama-3.2-1B-Instruct

| LLM | Approach | OBQA | ARC$_h$ | PIQA | CSQA | StrategyQA | QASC | SIQA | Avg. |
|---|---|---|---|---|---|---|---|---|---|
| Mistral (7B) | Fine-tuning | 87.2 | 74.5 | 88.4 | 82.9 | 72.9 | 81.5 | 79.8 | 81.0 |
| | Zero-shot | 72.6 | 72.1 | 75.7 | 70.4 | 42.8 | 65.9 | 69.6 | 67.0 |
| | KNOWPROXY | **81.0** | **73.8** | **84.3** | **74.5** | **69.4** | **75.8** | **77.4** | **76.6** |
| Llama 2 (13B) (4-bit quantized) | Fine-tuning | 84.6 | 73.8 | 87.8 | 82.0 | 70.7 | 78.6 | 81.7 | 79.9 |
| | Zero-shot | 61.0 | 56.3 | 72.6 | 53.3 | 55.0 | 52.9 | 48.6 | 57.1 |
| | KNOWPROXY | **75.0** | **63.9** | **81.3** | **75.4** | **71.2** | **71.3** | **75.6** | **73.4** |
| ChatGPT (gpt-3.5-turbo) | Zero-shot | 78.8 | 81.2 | 82.8 | 76.3 | 68.1 | 79.0 | 72.3 | 76.9 |
| | BBox-Adapter | 79.2 | 83.3 | **88.3** | 77.7 | 73.8 | 80.0 | 73.2 | 79.4 |
| | KNOWPROXY | **85.0** | **83.9** | 87.2 | **78.1** | **74.7** | **80.2** | **77.0** | **80.9** |

Table 2: Evaluation results for test accuracy (%) with diverse backbone LLMs. The best results are highlighted in **boldface**. Here, we use Llama-3.2 (1B) as the proxy.

| Model | Approach | OBQA | ARC$_h$ | PIQA | CSQA | StrategyQA | QASC | SIQA | Avg. |
|---|---|---|---|---|---|---|---|---|---|
| Llama 3.2 (3B) | Fine-tuning | 82.2 | 76.2 | 87.7 | 79.5 | 71.5 | 82.9 | 80.5 | 80.1 |
| | Zero-shot | 72.2 | 68.6 | 75.8 | 67.7 | 60.9 | 75.9 | 65.3 | 69.5 |
| w/ Llama 3.2 (1B) | | **80.2** | 75.2 | **83.4** | **75.0** | **72.9** | 78.1 | **76.3** | **77.3** |
| w/ LaMini-GPT (0.7B) | KNOWPROXY | 74.6 | 75.2 | 78.7 | 72.5 | 67.7 | 78.0 | 71.9 | 74.1 |
| w/ Qwen 2.5 (0.5B) | | 76.2 | **75.3** | 79.9 | 72.8 | 69.4 | **78.4** | 73.9 | 75.1 |

Table 3: Evaluation results for test accuracy (%) with diverse small language models. The best results are highlighted in **boldface**. Here, we use Llama-3.2 (3B) as the frozen LLM.

## 4.3 GENERAL APPLICABILITY TO ARCHITECTURES AND SCALES

**Applicability across LLM architectures and scales.** To demonstrate the broad applicability of KNOWPROXY (i.e., its model-agnostic nature), we evaluate its indirect fine-tuning effectiveness across diverse scenarios, including quantized LLMs (Llama-2 (13B))[10] and API-based black-box LLMs (ChatGPT). As shown in Table 2, KNOWPROXY consistently enhances performance across all benchmarks, regardless of the underlying LLM's capabilities. Notably, KNOWPROXY achieves significant performance gains even with black-box models (e.g., ChatGPT). These results highlight the plug-and-play nature of KNOWPROXY, which can effectively enhance both computationally demanding open-source models and black-box models with inaccessible internal parameters. By seamlessly integrating a smaller language model, KNOWPROXY provides a robust mechanism for indirect fine-tuning across a wide range of LLMs.

**Applicability across proxy choices.** We further analyze the general applicability of KNOW-PROXY using a diverse set of smaller language models, with the results presented in Table 3. The experimental results demonstrate that KNOWPROXY consistently enhances the zero-shot performance of LLMs across all smaller models. Moreover, we observe that the performance gains of LLMs are proportional to the capability of the smaller model (see Appendix C.4 for more details). Notably, through these results, along with additional analysis on the scalability of small models (Appendix C.5), KNOWPROXY demonstrates the practical adaptability of proxies, aligning with the characteristics of scaling laws (Kaplan et al., 2020) under environments restricted to small models.

## 4.4 COMPONENT ANALYSIS IN KNOWPROXY

**Ablation study.** We conduct an ablation study to assess the contribution of each component in KNOWPROXY to downstream task performance, with results summarized in Table 4. Our analysis considers the following variants: (i) **w/o routing**: Removes the dynamic routing mechanism, such that the proxy model is invoked for all inputs regardless of the LLM's

| Method | OBQA | PIQA | StrategyQA | SIQA |
|---|---|---|---|---|
| KNOWPROXY | **85.0** | **87.2** | **74.7** | **77.0** |
| w/o routing | 82.0 | 87.2 | 74.7 | 76.8 |
| w/o filtering | 85.0 | 86.2 | 72.1 | 76.0 |
| w/o adaptation | 80.6 | 85.1 | 59.4 | 75.3 |
| w/ LLM answer | 76.8 | 83.7 | 72.9 | 76.4 |

Table 4: Ablation of KNOWPROXY.

---

[10]We follow setup from (Dettmers et al., 2023).

confidence. (ii) **w/o filtering**: Omits the filtering process during knowledge generation, allowing all elicited knowledge to be used. (iii) **w/o adaptation**: Excludes the training of the proxy model on LLM-generated knowledge; instead, the proxy is trained only on the original inputs, and the generated knowledge is incorporated solely at inference time. (iv) **w/ LLM answer**: uses both the elicited knowledge and the LLM's prediction during training and inference of the proxy model.

The results demonstrate that removing the filtering and adaptation processes from KNOWPROXY substantially degrades performance on downstream tasks, underscoring the importance of both components. In particular, knowledge adaptation is critical, as its removal causes the largest performance drop (81.0 to 75.1 average). By contrast, excluding routing inference results in only a marginal decrease (81.0 to 80.2 average), indicating that our routing design—introduced primarily to enable efficient inference—does not compromise accuracy. This outcome confirms that the mechanism achieves its intended purpose: routing only the more difficult queries to the proxy, while easier cases are effectively handled by the LLM alone, thereby preserving accuracy while reducing inference overhead. Furthermore, KNOWPROXY with the LLM's predictions exhibits lower performance than the proposed method (81.0 to 77.5 average). These results suggest that LLM's predictions may introduce error propagation into the proxy, thereby negatively affecting the adaptation.

**Routing reliability.** We further analyze the reliability of the routing mechanism under the proposed uncertainty measure. For routing to be reliable in the proxy framework, the samples directed to the LLM (rather than the proxy) should be those that the LLM can solve easily. To examine this property, we evaluate the performance of the routed samples handled by the LLM. Specifically, we compare two uncertainty metrics: the previous approach (Tian et al., 2023), which relies on a single confidence score derived from the LLMs, and our proposed approach, which aggregates confidence scores across all generated knowledge. As shown in Figure 2, our method achieves higher performance as the confidence threshold increases, demonstrating its ability to distinguish easier cases that the LLM can solve reliably. In contrast, the baseline measure yields almost flat performance across different confidence thresholds, suggesting that a single prediction-level confidence does not provide meaningful guidance for routing. These results highlight that aggregating uncertainty across generated knowledge enables more effective confidence estimation and, in turn, more reliable routing decisions. To demonstrate the effectiveness of our aggregation approach, we further compare it with alternative aggregation methods, and the detailed results can be found in Appendix C.6.

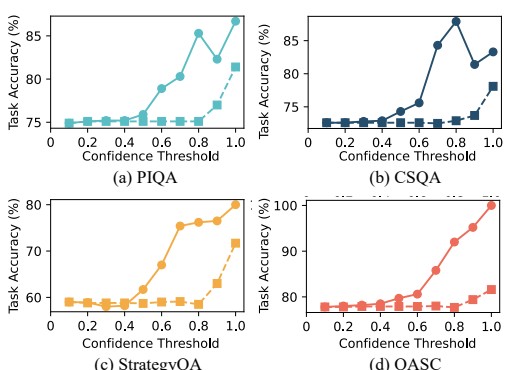

Figure 2: Confidence comparison between our method (solid lines) and baselines (dashed lines).

**Number of Knowledge.** Since the effectiveness of KNOWPROXY hinges on the amount of knowledge it elicits, we analyze how varying the number of generated knowledge instances influences performance. As shown in Figure 3, leveraging an appropriate amount of knowledge consistently improves results compared to the zero-shot LLM baseline, confirming its importance in adaptation. However, increasing the number of knowledge instances does not always yield additional gains and may even lead to performance degradation due to noisy or redundant reasoning, suggesting that generating a moderate number of knowledge pieces provides the most reliable improvements. Additional analysis on more powerful models can be found in Appendix C.6.

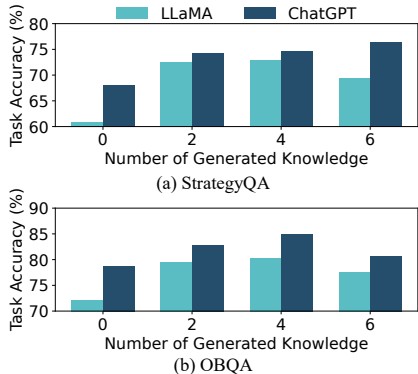

Figure 3: Performance across the different number of generated knowledge.

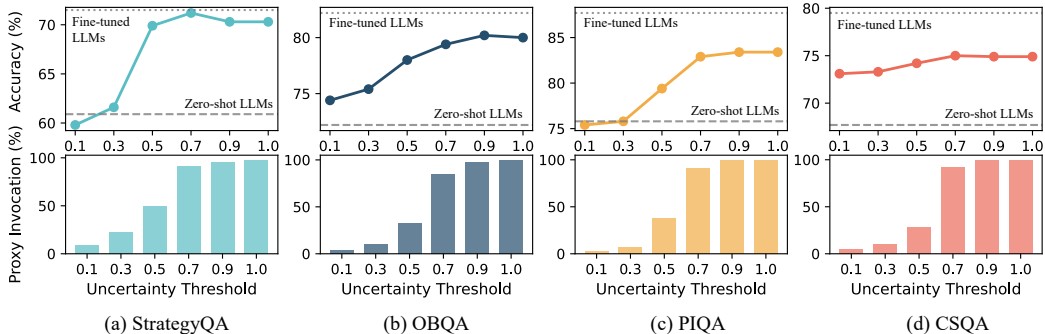

Figure 4: Comparison of inference cost and performance according to confidence thresholds.

| Method | Trained Knowledge | OBQA | ARC$_h$ | PIQA | CSQA | StrategyQA | QASC | Avg. |
|---|---|---|---|---|---|---|---|---|
| Zero-shot | - | 78.8 | 81.2 | 82.8 | 76.3 | 68.1 | 79.0 | 77.7 |
| | ChatGPT | 85.0 | 83.9 | 87.2 | 78.1 | 74.7 | 80.2 | 81.5 |
| KNOWPROXY | Llama 3.2 | 81.0 | 83.5 | 85.1 | 76.7 | 69.9 | 78.4 | 79.1 |
| | Mistral-v0.2 | 82.2 | 84.3 | 86.1 | 76.4 | 71.6 | 80.2 | 80.1 |

Table 5: Evaluation results for test accuracy (%). In these experiments, we train the small language models on different generated knowledge other than the target LLMs (i.e., ChatGPT).

## 4.5 TRADE-OFF OF PROXY ROUTING

To further validate the dynamic routing, we explore the balance between performance and the frequency of proxy model invocation in Figure 4. Unlike prior methods that rely on the small model for every input, KNOWPROXY selectively invokes the proxy only when needed. Our results show that, even with fewer proxy invocations, the framework maintains or improves performance across tasks, achieving accuracy levels competitive with direct fine-tuning of LLMs. This demonstrate that routing based on uncertainty not only improves efficiency by reducing unnecessary proxy usage but also preserves accuracy, underscoring the benefit of training proxies on LLM-generated knowledge rather than redistributing predictive distributions[11].

## 4.6 TRANSFERABILITY ACROSS DIVERSE LLMS

One notable advantage of KNOWPROXY is that it decouples the proxy model from the backbone LLM, allowing them to be swapped independently. This flexibility enables seamless replacement of the backbone LLM with newer models without requiring retraining of the proxy. To empirically validate this property, we conduct experiments by pairing a target LLM (ChatGPT) with proxy models that were trained on different LLMs. As shown in Table 5, these cross-trained proxies maintain strong performance when transferred to ChatGPT, despite being optimized with other LLMs. This demonstrates that the proxy learns to leverage elicited textual knowledge in a model-agnostic way, with language itself serving as the universal interface between LLMs and proxy models.

## 4.7 ANALYSIS OF REASONING GAINS FROM KNOWPROXY

KNOWPROXY adaptively combines the outputs of the LLM with those of the trained proxy model to generate the final reasoning. To examine how the proxy contributes beyond the LLM alone, we construct a confusion matrix based on the reasoning produced by the LLM and by KNOWPROXY (i.e., the LLM integrated with the fine-tuned proxy model). As shown in Figure 5, KNOWPROXY successfully corrects 38.7% of the questions that were originally answered incorrectly by the LLM. Notably, when applying KNOWPROXY, it achieves

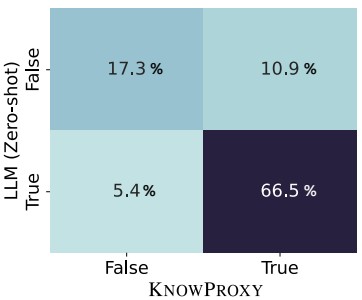

Figure 5: Confusion matrix from KNOWPROXY.

[11]Further analysis on memory efficiency is described in Appendix C.7.

a 5.5% higher rate of correct reasoning (10.9%) compared to incorrect reasoning (5.4%). These results highlight that KNOWPROXY effectively adapts LLMs by training a smaller model to leverage the elicited knowledge, thereby improving reasoning quality over the base LLM.

## 5 CONCLUSION

We have proposed KNOWPROXY, a novel framework for adapting LLMs through proxy models guided by elicited textual knowledge. Unlike prior approaches that have relied on predictive distributions, KNOWPROXY has leveraged knowledge expressed in natural language, making it broadly applicable across a wide range of LLMs, including black-box models that have only provided textual outputs. We have extensively evaluated the method across diverse benchmarks and training setups, including black-box LLMs and quantized LLMs of varying scales. The results have demonstrated that KNOWPROXY has consistently outperformed existing proxy-based approaches and even achieved performance comparable to direct fine-tuning. Our analysis has confirmed that the adaptive reasoning mechanism in KNOWPROXY has effectively balanced accuracy with efficiency, highlighting its promise as a promising alternative to direct fine-tuning.

### ACKNOWLEDGMENTS

This work was supported by the National Research Foundation of Korea (NRF) grant funded by the Korea government (MSIT) (No.RS-2025-00517221 and No.RS-2024-00415812) and Institute of Information & communications Technology Planning & Evaluation (IITP) grant funded by the Korea government (MSIT) (No.RS-2024-00439328, Karma: Towards Knowledge Augmentation for Complex Reasoning (SW Starlab), No.RS-2024-00457882, AI Research Hub Project, and No.RS-2019-II190079, Artificial Intelligence Graduate School Program (Korea University)).

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

## A    DETAILS OF KNOWLEDGE ADAPTATION

We define knowledge-augmented training datasets as $\mathcal{D}_{\mathcal{K}} = \{(x^i, k^i, y^i)\}_{i=0}^{N-1}$. Using the constructed dataset $\mathcal{D}_{\mathcal{K}}$, we train a small model $\mu_\phi(y \mid x, k)$ parameterized by $\phi$, to leverage the model-generated knowledge for the given target domains. The process by which the small language model predicts answers by leveraging knowledge from the large language model can be formulated as follows:

$$\pi'(y \mid x) \geq \mu_\phi(y \mid x, k)\,\pi_\theta(k \mid x), \tag{7}$$

$$\text{where} \quad \pi'(y \mid x) = \sum_{k_{all}} \mu_\phi(y \mid x, k_a)\,\pi_\theta(k_a \mid x), k_{all} \sim \pi_\theta(k|x).$$

By applying the empirical loss function on the dataset $D$, we can derive equation 8 as follows:

$$-\mathbb{E}_{\mathcal{D}}[\log \pi'(y \mid x)] \leq -\mathbb{E}_{\mathcal{D}}[\log \mu_\phi(y \mid x, k)] - \mathbb{E}_{\mathcal{D}}[\log \pi_\theta(k \mid x)]. \tag{8}$$

Therefore, our knowledge-guide objective function is as follows:

$$\min -\mathbb{E}_{\mathcal{D}}[\log \pi'(y \mid x)] \leq \min_\phi -\mathbb{E}_{\mathcal{D}}[\log \mu_\phi(y \mid x, k)] + \min_\theta -\mathbb{E}_{\mathcal{D}}[\log \pi_\theta(k \mid x)],$$

$$\text{where} \quad \pi'(y \mid x) = \sum_{k_{all}} \mu_\phi(y \mid x, k_a)\,\pi_\theta(k_a \mid x), k_{all} \sim \pi_\theta(k|x).$$

Here, based on previous studies (Prystawski et al., 2023; Gekhman et al., 2024) and empirical observations (see Table 4), we assume as follows:

$$\min_\phi -\mathbb{E}_{(x,y)\sim\mathcal{D}}\left[\log \mu_\phi(y \mid x, k')\right] \leq \min_\phi -\mathbb{E}_{(x,y)\sim\mathcal{D}}\left[\log \mu_\phi(y \mid x, k)\right],$$

$$\text{where} \quad k' = \{k_i \mid (k_i, c_i) \in \mathcal{K}, c_i > \alpha\}.$$

$$\therefore L_{\text{KNOWPROXY}} = \min_\phi -\mathbb{E}_{(x,y)\sim\mathcal{D}}\left[\log \mu_\phi(y \mid x, k')\right]. \tag{9}$$

## B    EXPERIMENTAL DETAILS

### B.1    DATASET DETAILS

| Dataset | Answer type | # of train data | # of test data |
|---|---|---|---|
| OpenBookQA | Multiple-choice | 4,957 | 500 |
| ARC-Challenge | Multiple-choice | 1,119 | 299 |
| CommonsenseQA | Multiple-choice | 9,741 | 1,221 |
| QASC | Multiple-choice | 8,134 | 926 |
| SocialIQA | Multiple-choice | 33,410 | 1,954 |
| ScienceQA | Multiple-choice | 2,000 | 500 |
| mCSQA (fr) | Multiple-choice | 8,047 | 1,005 |
| mCSQA (ru) | Multiple-choice | 6,623 | 827 |
| PhysicalIQA | Binary-choice | 16,113 | 1,838 |
| Winogrande | Binary-choice | 40,398 | 1,267 |
| BoolQ | Binary (True/False) | 9,427 | 3,270 |
| StrategyQA | Binary (True/False) | 2,061 | 229 |
| TruthfulQA | Open-ended text | 717 | 100 |

Table 6: Dataset descriptions and statistics.

Table 6 presents the key statistics of the datasets used in our experiments. Below, we provide a brief overview of each dataset.

**OpenBookQA** (Mihaylov et al., 2018) is a multiple-choice question-answering dataset on elementary science, designed to assess a model's commonsense knowledge.

**ARC-Challenge** (Clark et al., 2018) is a multiple-choice question-answering dataset consisting of scientific questions that are difficult to solve using either a retrieval-based algorithm or a word co-occurrence algorithm, designed to evaluate a model's complex reasoning ability.

**CommonsenseQA** (Talmor et al., 2019) is a multiple-choice question-answering dataset designed to evaluate a model's commonsense knowledge across common world scenarios.

**QASC** (Khot et al., 2020) is a multiple-choice question-answering dataset in grade school science, designed to evaluate the multi-hop reasoning ability of models.

**SocialIQA** (Sap et al., 2019) is a multiple-choice question-answering dataset designed to measure a model's social and emotional intelligence.

**ScienceQA** (Lu et al., 2022) is a multi-modal dataset designed for question-answering in the science domain, which is accompanied by annotated answers, lectures, and explanations. The dataset contains approximately 21,000 multi-modal questions. Following the experimental setting of (Sun et al., 2024), we exclude questions requiring image input and then randomly sample 2,000 questions for training and 500 for testing from the original train and test splits.

**mCSQA** (Sakai et al., 2024) is a multiple-choice question-answering dataset designed to evaluate the commonsense reasoning capabilities of models in diverse multilingual environments. In our experiments, we evaluate the multilingual commonsense reasoning abilities of language models in French and Russian settings.

**PhysicalIQA** (Bisk et al., 2020) is a binary question-answering dataset designed to evaluate a model's physical commonsense reasoning ability.

**Winogrande** (Sakaguchi et al., 2020) is a binary question-answering dataset designed to assess a model's commonsense knowledge by evaluating its ability to solve paired instances of coreference resolution.

**BoolQ** (Clark et al., 2019) is a binary question-answering dataset designed to assess a model's comprehensive reasoning ability using knowledge-intensive contexts and their associated questions.

**StrategyQA** (Geva et al., 2021) is a binary question-answering dataset designed to evaluate the ability of models to perform implicit multi-hop reasoning.

**TruthfulQA** (Lin et al., 2022) is a widely used generative dataset to evaluate a model's response quality in terms of truthfulness, factuality, and accuracy. The original dataset consists only of a test set. For evaluation, following the experimental setting of (Sun et al., 2024), we randomly select 100 samples to construct a test set and utilize the remaining samples as a train set.

## B.2 IMPLEMENTATION

| Hyperparameter | Value |
|---|---|
| Epoch | 6 |
| Batch size | 16 |
| Maximum input length | 512 |
| Optimizer | Adam |
| $\beta_1$ | 0.9 |
| $\beta_2$ | 0.999 |
| Learning rate | 2e-4 |
| Learning rate scheduling | Cosine decay |
| Weight decay | 0.0 |
| Warmup steps | 0.0 |
| LoRA rank | 64 |
| LoRA alpha | 8 |
| LoRA dropout | 0.1 |

Table 7: Hyperparameter settings for training.

We implement KNOWPROXY on Huggingface Transformers and PyTorch, and conduct all our experiments on two NVIDIA RTX A6000 GPUs. We fine-tune all models with LoRA in float32 mixed-precision, except for the Llama-2-13B-Chat model trained with QLoRA, and inference all open-sourced models in bfloat16 mixed-precision for efficiency. The hyperparameter settings for training KNOWPROXY and baselines are described in Table 7.

### B.3 KNOWLEDGE-ELICITING PROMPT TEMPLATE

We design a knowledge generation prompt template for each target task, taking into account the specific characteristics of each task. The detailed prompt template is provided below:

1) Classification benchmarks: Figure 8, Figure 9, Figure 10, Figure 11, and Figure 12.

2) TruthfulQA benchmark: Figure 13.

## C DETAILED EXPERIMENT RESULTS

### C.1 PERFORMANCE SENSITIVITY TO PROMPT DESIGNS

| Method | CSQA | QASC | StrategyQA |
|---|---|---|---|
| Zero-shot LLM | 67.7 | 75.9 | 60.9 |
| KNOWPROXY (Zero-shot) | 74.7 | 77.6 | 72.9 |
| KNOWPROXY (Few-shot) | 74.6 | 78.7 | 66.4 |
| KNOWPROXY (Decomposition) | 75.0 | 78.1 | 72.9 |

Table 8: Comparison of performance by prompt-induced knowledge quality.

We analyze the performance sensitivity of KNOWPROXY to prompt designs for eliciting knowledge from LLMs, since the quality of the extracted knowledge depends on the prompt design. Specifically, We compare the decomposition-style prompting strategy used in KNOWPROXY with two representative strategies (Liu et al., 2022a;b): zero-shot generation, which allows the LLM to freely generate knowledge without any examples, and few-shot generation, which provides the LLM with a small number of examples to specify the type of knowledge it should generate. As shown in Table 8, KNOWPROXY consistently improves performance compared to the LLM's zero-shot performance across prompting strategies. Moreover, we observe that prompting with decomposing strategy robustly achieves the highest performance across complex reasoning tasks.

### C.2 PERFORMANCE SENSITIVITY TO HYPER-PARAMETER

| $\alpha$ | OBQA | PIQA | QASC | StrategyQA |
|---|---|---|---|---|
| 0.0 | 76.8 | 79.2 | 74.4 | 70.7 |
| 0.1 | 74.8 | 80.1 | 73.2 | 65.5 |
| 0.2 | 76.0 | 80.1 | 73.7 | 67.7 |
| 0.3 | 76.4 | 80.6 | 74.7 | 70.3 |
| 0.4 | 76.8 | 79.7 | 75.9 | 67.7 |
| 0.5 | 77.4 | 80.1 | 75.6 | 69.0 |
| 0.6 | 77.8 | 79.3 | 73.9 | 70.3 |
| 0.7 | 73.8 | 79.4 | 75.4 | 73.8 |
| 0.8 | 74.4 | 80.1 | 74.2 | 66.8 |
| 0.9 | 72.6 | 79.1 | 71.2 | 66.8 |
| 1.0 | 64.0 | 74.4 | 62.9 | 62.4 |

Table 9: The detailed results of the knowledge filtering threshold on the proxy model's knowledge adaptation across complex reasoning tasks. We employ Llama 3.2 (3B) as LLM and Qwen 2.5 (0.5B) as SLM.

To investigate the impact of the knowledge filtering threshold during the proxy model training process, we evaluated the performance of KNOWPROXY, excluding its routing mechanism, across various complex reasoning benchmarks. As shown in Table 9, we observe that the optimal knowledge filtering threshold consistently ranges from 0.3 to 0.7 across all complex reasoning benchmarks. Specifically, the experiment demonstrates that excessively filtering LLM-generated knowledge diminishes the effectiveness of knowledge adaptation.

We also analyze the optimal routing threshold at which KNOWPROXY achieves its best performance across various LLMs, ranging from open-source to black-box models, and across multiple complex

| LLM | OBQA | ARC-challenge | PIQA | CSQA | StrategyQA | QASC | SIQA |
|---|---|---|---|---|---|---|---|
| Llama 3.2 (3B) | 1.0 | 0.3 | 1.0 | 0.7 | 0.6 | 0.5 | 1.0 |
| ChatGPT | 1.0 | 1.0 | 1.0 | 1.0 | 1.0 | 0.9 | 0.6 |
| Mistral (7B) | 1.0 | 0.8 | 1.0 | 0.8 | 1.0 | 1.0 | 1.0 |

Table 10: The detailed analysis of the optimal routing threshold at which KNOWPROXY achieves its best performance. Here, we employ Llama 3.2 (1B) as SLM.

| $\alpha$ | OBQA | PIQA | QASC |
|---|---|---|---|
| 0.0 | 76.8 | 79.2 | 78.7 |
| 0.1 | 74.8 | 80.1 | 78.2 |
| 0.2 | 76.0 | 80.1 | 77.6 |
| 0.3 | 76.4 | 80.6 | 78.3 |
| 0.4 | 76.8 | 79.7 | 78.9 |
| 0.5 | 77.4 | 80.1 | 78.5 |
| 0.6 | 77.8 | 79.3 | 78.8 |
| 0.7 | 73.8 | 79.4 | 78.3 |
| 0.8 | 74.4 | 80.1 | 78.5 |
| 0.9 | 72.6 | 79.1 | 76.9 |
| 1.0 | 64.0 | 74.4 | 78.1 |

Table 11: The detailed results of the knowledge filtering threshold on KNOWPROXY across complex reasoning tasks. Here. we employ Llama 3.2 (3B) as LLM and Qwen 2.5 (0.5B) as SLM.

reasoning benchmarks. As shown in Table 10, we observe that Llama 3.2 (3B) requires a relatively lower routing threshold compared to ChatGPT. Considering previous studies showing that Llama-series is relatively well-calibrated compared to ChatGPT (Xiong et al., 2024), this experiment implies that overconfident models may require a stricter routing threshold to effectively filter out unreliable predictions.

we further analyze the impact of the knowledge filtering threshold within KNOWPROXY. As shown in Table 11, the optimal knowledge filtering threshold for KNOWPROXY falls within the range of 0.3 to 0.6 across all complex reasoning benchmarks. Consistent with findings from previous work (Gekhman et al., 2024), these results suggest that excessive knowledge filtering can diminish the effectiveness of KNOWPROXY.

## C.3 DISTINCT QUESTION-ANSWERING TASKS

| Method | TruthfulQA (True + Info (%)) | ScienceQA (Acc. (%)) | mCSQA (fr) (Acc. (%)) | mCSQA (ru) (Acc. (%)) |
|---|---|---|---|---|
| Fine-tuning LLMs | 72.0 | 79.4 | 72.5 | 44.6 |
| Zero-shot LLMs | 66.0 | 77.6 | 64.7 | 36.6 |
| Fine-tuning SLMs | 61.0 | 66.6 | 65.7 | 36.2 |
| KNOWPROXY | 83.0 | 78.6 | 69.4 | 38.7 |

Table 12: Evaluation results on distinct tasks spanning truthful and informative (TruthfulQA), scientific (ScienceQA), and multilingual reasoning (mCSQA) domains. In these experiments, we employ Llama 3.2 (3B) as LLM and Llama 3.2 (1B) as SLM.

To assess the effectiveness of KNOWPROXY across diverse domains, we conduct evaluations on three representative benchmarks: truthfulness and informativeness (TruthfulQA), scientific reasoning (ScienceQA), and multilingual reasoning (mCSQA). As shown in Table 12, KNOWPROXY consistently outperforms the LLM's zero-shot performance across diverse domain-specific benchmarks. These results demonstrate the generalizability of KNOWPROXY across diverse domains.

| Model | Approach | OBQA | ARC$_h$ | PIQA | CSQA | StrategyQA | QASC | SIQA | Average |
|---|---|---|---|---|---|---|---|---|---|
| Llama 3.2 (3B) | Fine-tuning | 82.2 | 76.2 | 87.7 | 79.5 | 71.5 | 82.9 | 80.5 | 80.1 |
| | Zero-shot | 72.2 | 68.6 | 75.8 | 67.7 | 60.9 | 75.9 | 65.3 | 69.5 |
| w/ Llama 3.2 (1B) | Fine-tuning | 73.2 | 60.9 | 80.3 | 72.0 | 66.5 | 68.0 | 74.9 | 70.8 |
| | KNOWPROXY | 80.2 | 75.2 | 83.4 | 75.0 | 72.9 | 78.1 | 76.3 | 77.3 |
| w/ LaMini-GPT (0.7B) | Fine-tuning | 56.0 | 27.3 | 70.4 | 49.5 | 61.9 | 20.4 | 68.2 | 50.5 |
| | KNOWPROXY | 74.6 | 75.2 | 78.7 | 72.5 | 67.7 | 78.0 | 71.9 | 74.1 |
| w/ Qwen 2.5 (0.5B) | Fine-tuning | 68.8 | 43.9 | 73.9 | 65.4 | 59.7 | 64.2 | 69.1 | 63.6 |
| | KNOWPROXY | 76.2 | 75.3 | 79.9 | 72.8 | 69.4 | 78.4 | 73.9 | 75.1 |

Table 13: The detailed results on applicability across proxy models. In these experiments, we employ Llama 3.2 (3B) as the frozen LLM.

### C.4 DETAILED RESULTS ON APPLICABILITY ACROSS PROXY CHOICES

We further analyze how the capability of the proxy model influences the performance of KNOW-PROXY. As shown in Table 13, we observe that the performance of KNOWPROXY is influenced by the capability of the lightweight proxy model. These results demonstrate that the capability of the proxy model plays an important role in enhancing the performance of KNOWPROXY. Notably, even when employing lower-capability proxy models such as LaMini-GPT (0.7B), KNOWPROXY consistently achieves competitive performance across diverse benchmarks. Together, these results underscore the broad applicability of KNOWPROXY across proxy choices.

### C.5 PROXY MODEL SCALABILITY

To examine the relationship between the scalability of proxy models and KNOWPROXY, we evaluate KNOWPROXY across various reasoning benchmarks using the Pythia family, which differs only in model size while maintaining consistent design choices and training processes. As shown in Figure 6, we observe that the performance of KNOWPROXY increases as the the proxy size increases. These results suggest that the effectiveness of KNOW-PROXY scales with proxy model size, consistent with the characteristics of scaling laws (Kaplan et al., 2020) under settings restricted to small models.

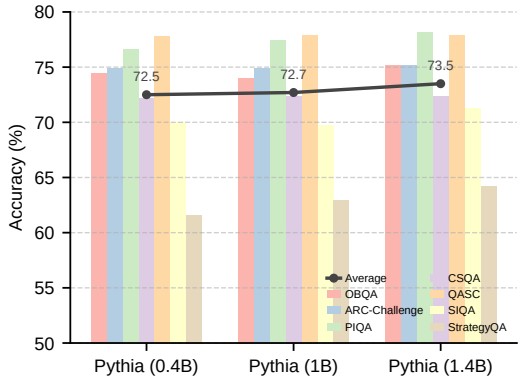

Figure 6: Comparison of effectiveness with respect to the scalability of small models.

### C.6 ADDITIONAL COMPONENT ANALYSIS IN KNOWPROXY

| Method (KNOWPROXY) | OBQA | ARC$_h$ | StrategyQA | QASC |
|---|---|---|---|---|
| Arithmetic mean | 78.2 | 75.5 | 72.5 | 77.8 |
| Geometric mean | 79.6 | 75.5 | 70.3 | 77.8 |
| Product (ours) | 80.2 | 75.5 | 72.9 | 78.1 |

Table 14: Confidence aggregation method comparison.

**Routing reliability.** We also analyze how our confidence aggregation method effectively determines the routing path compared to alternative aggregation schemes. Specifically, we compare the performance of KNOWPROXY using our confidence aggregation method with its performance under alternative aggregation methods, including the arithmetic mean, which is a representative weighted sum, and the geometric mean, which is a representative weighted product. As shown in Table 14, our method consistently achieves the highest performance across multiple complex reasoning benchmarks. These results highlight that selecting an aggregation method that effectively mitigates the penalties caused by the overconfidence of LLMs is crucial for achieving the additional performance gains of KNOWPROXY.

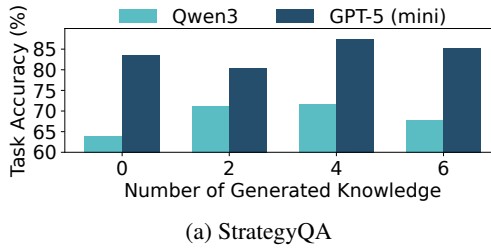 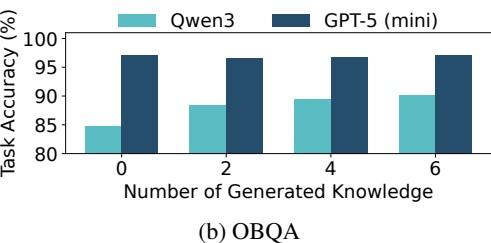

(a) StrategyQA            (b) OBQA

Figure 7: Performance of more powerful models across different numbers of generated knowledge.

**Number of Knowledge.**    To further validate the necessity of generating an appropriate amount of knowledge, we conduct additional analysis on newer and more powerful models (e.g., Qwen3 (4B) and GPT-5 (Mini)). As shown in Figure 7, consistent with the results observed for Llama-3.2-3B-Instruct and GPT-3.5 Turbo in Figure 3, we observe that the necessity of generating an appropriate amount of knowledge remains crucial for maximizing KNOWPROXY's performance, even in these stronger models.

## C.7    RESOURCE ANALYSIS

| Approach | PIQA | SIQA | OBQA | QASC |
|---|---|---|---|---|
| Direct Fine-tuning (LoRA) | 74.5 | 38.0 | 30.4 | 36.7 |
| Proxy-tuning | 33.9 | 16.9 | 15.5 | 16.1 |
| CombLM | 40.4 | 23.4 | 22.0 | 22.6 |
| KNOWPROXY (Llama 3.2 (1B)) | 33.9 | 24.6 | 26.9 | 27.1 |
| KNOWPROXY (LaMini-GPT (0.7B)) | 38.1 | 26.8 | 29.2 | 31.2 |
| KNOWPROXY (Qwen 2.5 (0.5B)) | 24.6 | 16.8 | 19.6 | 19.3 |

Table 15: Comparison of the maximum GPU memory required for training process. We set the target LLM for training in each task to Llama 3.2 (3B). For each proxy model, KNOWPROXY is evaluated under the hyperparameter settings that yield the best performance.

We further conduct a resource analysis to demonstrate the efficiency of KNOWPROXY in terms of the resources required during training. To further highlight the contribution of KNOWPROXY, which independently trains smaller proxy models while leveraging textual knowledge, we analyze how textual knowledge influences resource burden for training. As shown in Table 15, we observe that KNOWPROXY, across all proxy choices, is consistently more memory-efficient than parameter-efficient methods such as LoRA during training. Notably, on PhysicalIQA with long input sequences, KNOWPROXY exhibits substantial memory efficiency comparable to existing proxy-based approaches that fine-tune only the smaller model in target domains. These results demonstrate that KNOWPROXY is a resource-efficient approach for effectively adapting LLMs to target domains, ranging from white-box to black-box settings.

## D    FUTURE WORK

While we demonstrate that KNOWPROXY achieves performance comparable to direct fine-tuning through extensive experimental validation, there remain promising directions for further improvement and more in-depth analysis.

**Overconfidence issue of LLMs.**    When determining the reasoning path to reduce inference overhead while maintaining performance, KNOWPROXY utilizes the LLM's confidence scores. However, we empirically observe that LLMs tend to exhibit overconfidence. To mitigate this issue, we comprehensively consider the confidence scores of both the generated knowledge and the prediction, and propose empirically determined routing thresholds. Nevertheless, further improving the reliability and calibration of LLM-elicited confidence scores remains an important direction for further enhancing the effectiveness of KNOWPROXY.

**Discussion on the proxy model.** Recent works (Gekhman et al., 2024; Wei et al., 2025; Akter et al., 2026) suggest that fine-tuning primarily adapts models to task- or domain-specific distributions rather than fundamentally injecting new knowledge into pre-trained models. In light of this perspective, the LLM-agnostic behavior of KNOWPROXY may not simply stem from transferring the LLM's internal textual knowledge into the proxy model. Instead, it may arise from training the proxy model to adapt its reasoning to the structure and content of elicited knowledge, as well as to perform domain adaptation to the target tasks. Further empirical and theoretical analysis of the underlying mechanisms of this behavior remains a promising direction for future work.

## E  THE USE OF LARGE LANGUAGE MODELS (LLMS)

Large language models (LLMs) were employed solely to enhance the clarity and grammar of the manuscript, such as grammar correction and expression refinement. The LLMs were not used to generate scientific insights, analyze results, or draw conclusions.

## Qwen3 (4B)

You are a helpful chatbot.
Your task is to extract the knowledge needed to easily solve the given question and then predict the correct answer using your generated knowledge.
Specifically, the knowledge you extract must be relevant to what you believe is the correct answer.
You should correctly generate the knowledge.

### Question
{Specific Question Here}

Given all of the above, read the question, break down the problem into 4 steps which are knowledges needed to easily solve the given problem, think step by step, give your confidence in each step. Note that the confidence indicates how likely you think your answer is true.

# Please use the following format to answer.
{Specific Output Format Here}
# Generate all requested items without omission, strictly following the given format.
# Do not generate overly verbose knowledge.

The Response is (

Figure 8: The designed prompt for Qwen3 (4B) aimed at generating knowledge in complex reasoning benchmarks.

## Llama 3.2 (3B) & Llama 2 (13B)

You are a helpful assistant that extracts the knowledge needed to easily solve the given question and then predicts the correct answer.

# Guidelines
- Your task is to extract the knowledge needed to easily solve the given question and then predict the correct answer using your generated knowledge.
- Specifically, the knowledge you extract must be relevant to what you believe is the correct answer.

# CAUTION
- You MUST NOT respond any of the given answer choices as the answer.
- Correctly generate the knowledge.
- Strictly follow the answer format mentioned.
- Do not overestimate your confidence.

# Question
{Specific Question Here}
Human: Given all of the above, read the question, break down the problem into 4 steps which are knowledge needed to easily solve the given problem, think step by step, give your confidence in each step. Note: The confidence indicates how likely you think your answer is true.

Use the following format to answer:
{Specific Output Format Here}
Assistant: The Response is (

Figure 9: The designed prompt for Llama family aimed at generating knowledge in complex reasoning benchmarks.

## Mistral (7B)

You are a helpful chatbot.
Your task is to extract the knowledge needed to easily solve the given question and then predict the correct answer using your generated knowledge.
Specifically, the knowledge you extract must be relevant to what you believe is the correct answer.
You should correctly generate the knowledge.

### Question
{Specific Question Here}

Given all of the above, read the question, break down the problem into 4 steps which are knowledges needed to easily solve the given problem, think step by step, give your confidence in each step. Note that the confidence indicates how likely you think your answer is true.

# Please use the following format to answer.
{Specific Output Format Here}
# Generate all requested items without omission, strictly following the given format.
# Do not generate overly verbose knowledge.

The Response is (

Figure 10: The designed prompt for Mistral (7B) aimed at generating knowledge in complex reasoning benchmarks.

## ChatGPT (GPT-3.5-turbo) & GPT-5 (mini)

You are a helpful assistant that extracts the knowledge needed to easily solve the given question and then predicts the correct answer.

## Guidelines
1. Your task is to extract the knowledge needed to easily solve the given question and then predict the correct answer using your generated knowledge.
2. Specifically, the knowledge you extract must be relevant to what you believe is the correct answer.
3. Given all of the above, read the question, break down the problem into 4 steps which are knowledge needed to easily solve the given problem, think step by step, give your confidence in each step.
4. Note: The confidence indicates how likely you think your answer is true.
5. Use the following format to answer:
{Specific Output Format Here}

## CAUTION
- You MUST NOT respond any of the given answer choices as the answer.
- Correctly generate the knowledge.
- Do not overestimate your confidence.

# Question
{Specific Question Here}

Figure 11: The designed prompt for API-based models (e.g., ChatGPT (GPT-3.5-turbo) and GPT-5 (mini)) aimed at generating knowledge in complex reasoning benchmarks.

---

### Llama 3.2 (3B) (ScienceQA)

You are a helpful assistant that extracts the knowledge needed to easily solve the given question and then predicts the correct answer.

# Guidelines
- Your task is to extract the knowledge needed to easily solve the given question and then predict the correct answer using your generated knowledge.
- Specifically, the knowledge you extract must be relevant to what you believe is the correct answer.

# CAUTION
- You MUST NOT respond any of the given answer choices as the answer.
- Correctly generate the knowledge.
- Strictly follow the answer format mentioned.
- Do not overestimate your confidence.

# Question
{Specific Question Here}
Human: Given all of the above, read the question, break down the problem into 4 steps which are knowledge needed to easily solve the given problem, think step by step, give your confidence in each step. Note: The confidence indicates how likely you think your answer is true.

Use the following format to answer:
{Specific Output Format Here}
Assistant: The Response is (

---

Figure 12: The designed prompt for Llama 3.2 aimed at generating knowledge associated with the ScienceQA dataset.

---

### Llama 3.2 (3B) (TruthfulQA)

You are a helpful, respectful, and honest assistant that extracts the knowledge needed to solve the question easily and helpfully, and then predicts the correct and safe answer.

# Guidelines
- Your task is to extract the knowledge required to solve the given question easily and helpfully, and then predict the correct and safe answer based on your generated knowledge.
- Specifically, the knowledge you extract must be relevant to what you believe is the correct and safe answer.
- Your responses should not include any harmful, unethical, racist, sexist, toxic, dangerous, or illegal content.
- If the given question does not make sense or is not factually coherent, please don't answer false information.

# CAUTION
- Correctly generate the knowledge.
- Please ensure that your knowledge and answer are socially unbiased and positive in nature.
- Strictly follow the answer format mentioned.
- Do not overestimate your confidence.

# Question
{Specific Question Here}
Human: Given all of the above, read the question, break down the problem into 4 steps which are knowledge needed to solve the given problem easily and helpfully, think step by step, give your confidence in each step. Note: The confidence indicates how likely you think your answer is true.

Use the following format to answer:
{Specific Output Format Here}
Assistant: The Response is (

---

Figure 13: The designed prompt for Llama 3.2 aimed at generating knowledge associated with TruthfulQA dataset.

