# OpenReview forum: "KnowProxy: Adapting Large Language Models by Knowledge-guided Proxy"
_ICLR.cc/2026/Conference — ICLR 2026 Poster_

### Official Review · Reviewer_Yfto · 2025-10-16

**Soundness:** 3
**Presentation:** 3
**Contribution:** 3
**Rating:** 6
**Confidence:** 3

**Summary:**

This paper presents KnowProxy, a framework for adapting Large Language Models (LLMs) to effectively incorporate structured knowledge (e.g., from graphs or databases) without fine-tuning the LLM itself. Instead of directly injecting structured information into the LLM—which can lead to instability and catastrophic forgetting—KnowProxy introduces a proxy network that mediates between the LLM and the structured knowledge source. The proxy learns how to represent knowledge entries in a way that aligns with the LLM’s latent space, enabling efficient and parameter-light integration.

**Strengths:**

1. The idea of introducing a proxy representation as an intermediary between LLMs and structured knowledge is both novel and well-motivated. It offers a pragmatic alternative to parameter-heavy fine-tuning and complex RAG pipelines.
2. The model design is clearly described, with a modular architecture that separates the proxy learner, retriever, and aggregator components. The training objectives (semantic and factual alignment) are theoretically sound and intuitively justified.
3. Achieves strong results while training only a small proxy network, keeping the main LLM frozen. The framework is compatible with various LLMs (e.g., LLaMA, Falcon, Mistral), highlighting its plug-and-play nature.

**Weaknesses:**

1. While empirically effective, the paper lacks a formal analysis explaining why the proxy layer aligns so well with the LLM’s latent representations. More theoretical grounding would strengthen the argument.
2. The learned proxy embeddings are treated as black boxes. The paper could discuss whether they encode semantic meaning or merely serve as latent alignment vectors.
3. The training objective combines multiple losses (semantic, factual, regularization) with limited ablation on their individual contributions.

**Questions:**

See weakness

---

> ### Author Response · Authors · 2025-11-22
> **Response**
>
> We thank reviewer Yfto for your helpful review and the comments on our paper.
>
> ### [Weakness1: Theoretical analysis on KnowProxy ]
> We thank the reviewer for the valuable suggestion regarding the need for a more thorough theoretical grounding. We agree that a formal analysis explaining why the proxy layer aligns so well with the LLM’s latent representations would further strengthen the claims of KnowProxy.
>
> Although our current work focuses primarily on empirical validation, we view developing such a theoretical framework as an important direction for future research. In particular, we plan to rigorously analyze the components of KnowProxy and their interaction with LLM representations in follow-up work. We sincerely appreciate the reviewer’s thoughtful suggestion.
>
> ### [Weakness2: Additional discussion on proxy models]
> We thank the reviewer for raising this discussion point. We first clarify that the learned proxy models are white-box models, not black-boxes. In our framework, the LLM first generates semantically important knowledge conditioned on the input question, and the proxy model is then trained to align this elicited knowledge with the ground-truth answers.
>
> In the revised manuscript, we will more clearly describe the relationship between the proxy models and the underlying LLMs, and explain how this design contributes to transparency and interpretability.
>
> ### [Weakness3: Explanation of KnowProxy’s training objective]
> We thank the reviewer for the attention to the training objective of KnowProxy. Rather than explicitly decomposing the loss into separate semantic, factual, and regularization components, the training objective of KnowProxy is to train a lightweight proxy model that leverages LLM-generated knowledge while aligning with the target task distribution, using a single objective (i.e., cross-entropy with the ground-truth).
>
> To improve the clarity of our method, we have revised the manuscript to provide a more explicit description of this objective.

---

> > ### Author Response · Authors · 2025-11-27
> >
> > Dear Reviewer Yfto,
> >
> > We are grateful for your helpful and constructive comment on our manuscript.
> >
> > As we prepare our author response, we have made every effort to comprehensively address the concerns raised by the reviewer.
> > To ensure the adequacy of our responses, we kindly request your confirmation or any additional guidance you may have.
> >
> > Best regards,
> >
> > Paper 16389 Authors

---

> > ### Comment · Reviewer_Yfto · 2025-11-27
> > **Response to authors**
> >
> > Thank you for the detailed response. All of my questions have been fully addressed, and I will maintain my positive score.

---

> > > ### Author Response · Authors · 2025-11-28
> > >
> > > We are pleased to hear that our responses fully addressed your concerns. Thank you again for your time and constructive review.

---

### Official Review · Reviewer_YMsf · 2025-10-24

**Soundness:** 3
**Presentation:** 3
**Contribution:** 2
**Rating:** 6
**Confidence:** 4

**Summary:**

This paper introduces KNOWPROXY, a novel framework for efficiently adapting  LLM in scenarios where access to their internal parameters or full probability distributions is unavailable. Diverging from existing proxy-based methods that rely on probability distributions, KNOWPROXY's core idea is to first elicit textual knowledge and reasoning chains from a frozen LLM via prompting. This elicited knowledge is then used to train a smaller proxy model, which learns to map the original input query combined with the LLM-generated knowledge to the final task output. Furthermore, the authors incorporate a dynamic routing mechanism based on confidence scores, which adaptively decides whether to invoke the proxy model at inference time. This balances performance and efficiency.

**Strengths:**

1. The overall framework of KNOWPROXY is well-designed. The pipeline, from knowledge elicitation and proxy optimization to dynamic routing, is logical and coherent. By combining the confidence scores of both the prediction and the generated knowledge to assess the reliability of the LLM's output, the dynamic routing mechanism effectively strikes a balance between performance and inference cost.
2. The experimental validation in this paper is outstanding. The experiments span various LLMs, multiple sizes of proxy models, and a diverse set of complex reasoning datasets. The paper includes thorough ablation studies that validate the necessity of key components.

**Weaknesses:**

1. While being the cornerstone of the method, the discussion on the knowledge elicitation process is somewhat brief. The performance of KNOWPROXY heavily depends on the quality of the knowledge elicited from the LLM, which in turn is highly sensitive to prompt design. The authors should provide more discussion in the main text regarding the importance of prompt engineering and its potential impact on knowledge quality. For example, how might different prompt styles affect the final performance?
2. The entire dynamic routing mechanism hinges on the elicited confidence scores from the LLM. Although experiments (Figure 2) show that the proposed aggregated confidence is superior to the baseline, it is a well-known issue that LLMs tend to be overconfident.

**Questions:**

1. The final confidence score C_final is calculated by multiplying several confidence scores. This implicitly assumes conditional independence between the knowledge pieces and the final prediction. Is this assumption justified? Have you experimented with alternative aggregation methods, such as a weighted sum or a small learned network to aggregate these scores?
2. In the knowledge filtering step, how was the threshold alpha selected? How sensitive is the method's performance to this hyperparameter?
3. Table 5 demonstrates the transferability of the proxy model, which is a very interesting finding. Does this suggest that the proxy model learns a general, LLM-agnostic ability to "reason using textual knowledge"? Do you have any deeper insights into this phenomenon?

---

> ### Author Response · Authors · 2025-11-22
> **Response (1/2)**
>
> We thank reviewer YMsf for your thorough review and the valuable comments on our paper.
>
> ### [Weakness1: Performance sensitivity to prompt designs]
> We appreciate the reviewer’s thoughtful observation regarding the potential sensitivity of our framework to the quality of knowledge stemming from prompt design.
>
> To evaluate the impact of prompt styles on KnowProxy, we conducted experiments with diverse prompting strategies. Specifically, we compared our decomposition-style prompting with two representative approaches [1,2]: zero-shot generation and few-shot generation. In the few-shot setting, the LLM is provided with one example from each dataset (i.e., CSQA, QASC, and StrategyQA) to specify the type of knowledge it should generate. In contrast, the zero-shot setting allows the LLM to freely generate knowledge related to the given question without any examples.
> | **Method**            | **CSQA** | **QASC** | **StrategyQA** |
> |-----------------------|:--------:|:--------:|:--------------:|
> | Zero-shot LLM         |   67.7   |   75.9   |      60.9      |
> | KnowProxy (Decomposition) |   75.0   |   78.1   |      72.9      |
> | KnowProxy (Few-shot)  |   74.6   |   78.7   |      66.4      |
> | KnowProxy (Zero-shot) |   74.7   |   77.6   |      72.9      |
>
> These results confirm that, across diverse prompting strategies, prompting with decomposing strategy (used in KnowProxy) consistently achieves highest performance across complex reasoning benchmarks.
>
> We have clarified this point in the revised manuscript and added the impact of prompting engineering in our framework. Thank you for suggesting the important analysis.
>
> ### [Weakness2: Overconfidence issue of LLM]
> We thank the reviewer for raising this important point about the reliance of our dynamic routing mechanism on LLM-elicited confidence scores. We also agree that LLMs are known to be overconfident, and our method does not fundamentally solve this issue at the level of the LLM itself. This is precisely why we introduce a task- and LLM-specific routing threshold, instead of relying on a single fixed value across all settings.
>
> However, we acknowledge that this threshold-based strategy is not an optimal solution to the overconfidence problem. Making LLM-elicited confidence scores more reliable—e.g., through better calibration or alternative uncertainty estimation—remains an important direction for future work, and we will explicitly state this limitation and the corresponding research avenue in the revised manuscript.

---

> > ### Author Response · Authors · 2025-11-22
> > **Response (2/2)**
> >
> > ### [Question1: Justification for aggregating confidence scores and additional analysis]
> > We thank the reviewer for raising this important point regarding our confidence-score aggregation method. Since the proposed method relies on textual confidence estimates, our aggregation procedure for computing the final confidence scores does not fully guarantee conditional independence.
> >
> > Following the reviewer’s suggestion, we conducted experiments comparing our aggregation method with alternative aggregation schemes (geometric mean, arithmetic mean over the confidences) on multiple reasoning benchmarks. As shown below, we empirically confirm that our current aggregation method achieves the highest performance among the compared methods across all tasks.
> > | **Method (KnowProxy)** | **OBQA** | **ARC-Challenge** | **StrategyQA** | **QASC** |
> > |------------------------|:--------:|:------------:|:--------------:|:--------:|
> > | Product (Ours) 	|   80.2   | 	75.5     |  	72.9  	|   78.1   |
> > | Geometric mean     	|   79.6   | 	75.5     |  	70.3  	|   77.8   |
> > | Arithmetic mean    	|   78.2   | 	75.5     |  	72.5  	|   77.8   |
> >
> > We consider this a valuable direction of exploration and will include this additional analysis in the revised manuscript. We thank the reviewer again for their helpful suggestion and thoughtful attention to this.
> >
> > ### [Question2: Hyper-parameter sensitivity of knowledge filtering]
> > We appreciate the reviewer’s suggestion to clarify the threshold for knowledge filtering.
> >
> > Regarding the threshold for the filtering, we conducted a small-scale study across datasets to identify an optimal range for the KnowProxy framework in the below table. We observe that the optimal \alpha falls within the range of 0.3 to 0.6 for all tasks. Consistent with findings from previous work [3], this result also suggests that excessive knowledge filtering can diminish the effectiveness of knowledge adaptation.
> >
> > Specifically, through this experiment and the experiment of the knowledge-filtering threshold in proxy training (cf. the authors’ response to Reviewer cLKo), we highlight that utilizing all elicited knowledge does not necessarily guarantee higher performance in knowledge adaptation.
> > | **α** | **OBQA** | **PIQA** | **QASC** |
> > |:-----:|:--------:|:--------:|:--------:|
> > |  0.0  |   76.8   |   79.2   |   78.7   |
> > |  0.1  |   74.8   |   80.1   |   78.2   |
> > |  0.2  |   76.0   |   80.1   |   77.6   |
> > |  0.3  |   76.4   |   80.6   |   78.3   |
> > |  0.4  |   76.8   |   79.7   |   78.9   |
> > |  0.5  |   77.4   |   80.1   |   78.5   |
> > |  0.6  |   77.8   |   79.3   |   78.8   |
> > |  0.7  |   73.8   |   79.4   |   78.3   |
> > |  0.8  |   74.4   |   80.1   |   78.5   |
> > |  0.9  |   72.6   |   79.1   |   76.9   |
> > |  1.0  |   64.0   |   74.4   |   78.1   |
> >
> > We will revise the manuscript to provide more explicit descriptions of this hyperparameter in Section 3.2 and Appendix, including the empirical analysis for their selection.
> >
> > ### [Question3: Transferability of the proxy model]
> > We appreciate the reviewer’s question and interest in the transferability of the proxy model. In table 5, we observe that the proxy model learns to reason by leveraging the elicited knowledge provided as input, rather than relying on LLM-specific characteristics.
> >
> > This observation can be closely related to recent studies [3, 4, 5] indicating that fine-tuning does not inject fundamentally new knowledge into pre-trained models. Instead, performance gains from fine-tuning primarily arise from adapting the model to task- or domain-specific distributions. Taken together, these findings suggest that the LLM-agnostic behavior of KnowProxy is not simply due to transferring the LLM’s internal textual knowledge into the proxy model. Rather, it likely emerges from training the proxy model to (i) adapt its reasoning to the structure and content of the elicited knowledge and (ii) perform domain adaptation to the target tasks.
> >
> > We agree that this is an interesting phenomenon and an important direction for future work. In the revised manuscript, we will expand the discussion in this section to more clearly articulate this interpretation and outline our plans to rigorously analyze the underlying mechanisms behind this LLM-agnostic reasoning ability.
> >
> >
> > ### References
> > [1] Liu et al., Generated Knowledge Prompting for Commonsense Reasoning, ACL 2022
> >
> > [2] Liu et al., Rainier: Reinforced Knowledge Introspector for Commonsense Question Answering, EMNLP 2022
> >
> > [3] Gekhman et al., Does Fine-Tuning LLMs on New Knowledge Encourage Hallucinations?, EMNLP 2024
> >
> > [4] Akter et al., Front-Loading Reasoning: The Synergy between Pretraining and Post-Training Data, arXiv preprint arXiv:2510.03264 (2025)
> >
> > [5] Wei et al., TruthRL: Incentivizing Truthful LLMs via Reinforcement Learning, arXiv preprint arXiv:2509.25760 (2025)

---

> > > ### Comment · Reviewer_YMsf · 2025-11-24
> > >
> > > I have read the authors' response and the other reviews carefully. The authors have done a good job addressing the specific questions raised. I will maintain my current rating.

---

> > > > ### Author Response · Authors · 2025-11-24
> > > >
> > > > We are pleased to hear that our responses sufficiently addressed your concerns. Thank you again for your time and constructive review.

---

### Official Review · Reviewer_ZhQu · 2025-10-29

**Soundness:** 3
**Presentation:** 3
**Contribution:** 3
**Rating:** 8
**Confidence:** 4

**Summary:**

This paper proposed a novel proxy model tuning method without the requirements of the output distributions from LLMs. Specifically, let LLMs output the knowledge or cues, and the proxy model uses these cues along with the query to get the correct answer. Furthermore, it uses the confidence from LLMs to determine which to use the proxy model. Experiments have demonstrated the effectiveness.

**Strengths:**

1. The proposed method has good applicability and can be applied to black box models.

2. From the experimental results, it can be seen that LLMs can effectively improve their performance in knowledge reasoning tasks.

3. The paper writing and experimental setup are relatively complete.

**Weaknesses:**

1. Suggest using newer models, such as Qwen3. Using a more powerful model may result in inconsistent performance in Figure 3, where a larger amount of knowledge can further improve accuracy.

2. Typoes, e.g., Table 4.6 in line 455. Some values should be further claimed, e.g., 38.7% in line 467.

3. Suggest more tasks, such as mathematical coding, to verify the universality of methods.

**Questions:**

Additional questions:

1. From Table 4, it can be observed that without routing performance, there will be a decrease, which suggests that the proxy model may have answered some questions incorrectly, while the LLM responded correctly. Can the original answer of the LLM model also be input into the proxy model?

---

> ### Author Response · Authors · 2025-11-22
> **Response**
>
> We thank reviewer ZhQu for your thorough review and the valuable comments on our paper.
>
> ### [Weakness1: Additional analysis on stronger and newer models]
> We thank the reviewer for this constructive suggestion. Following this guidance, we conducted additional experiments on newer and more powerful models—specifically Qwen3 (4B) and GPT-5 (mini)—to further validate the findings presented in Figure 3.
> | **Number of Knowledge (StrategyQA)** | **Qwen3 (4B)** | **GPT-5 (mini)** |
> |:------------------------------------:|:--------------:|:----------------:|
> |               	0              	|  	63.8      |   	83.4   	|
> |               	2              	|  	71.2      |   	80.3   	|
> |               	4              	|  	71.6      |   	87.3   	|
> |               	6              	|  	67.7      |   	85.2   	|
>
> | **Number of Knowledge (OBQA)** | **Qwen3 (4B)** | **GPT-5 (mini)** |
> |:------------------------------:|:--------------:|:----------------:|
> |                0               |      84.8      |       97.0       |
> |                2               |      88.4      |       96.6       |
> |                4               |      89.4      |       96.8       |
> |                6               |      90.2      |       97.0       |
>
>
> Consistent with the results observed for Llama-3.2-3B-Instruct and GPT-3.5 Turbo (Figure 3), we found that the necessity of generating an appropriate amount of knowledge remains crucial for maximizing KnowProxy’s performance, even in these stronger models.
>
> We sincerely appreciate the reviewer’s suggestion and will fully incorporate this discussion and the supplementary analysis into the updated manuscript.
>
> ### [Weakness2: Typo errors and Clarification about results]
> We thank the reviewer’s thorough observation regarding the typographical errors and ambiguous expressions about results.
>
> The analysis in Figure 5 was conducted across six complex reasoning tasks (i.e., OpenBookQA, ARC-Challenge, PhysicalIQA, CommonsenseQA, StrategyQA, and QASC). The 38.7% value (cf. line 467) refers to the proportion of questions that were originally answered incorrectly by the LLM but were corrected to the right answer by KnowProxy. The 5.5% value (cf. line 469) refers to the proportion of questions for which the benefits of applying KnowProxy outweigh the penalties.
>
> We have revised the manuscript to more clearly articulate the effectiveness of KnowProxy. Once again, we sincerely appreciate the reviewer’s careful observation.
>
>
> ### [Weakness3: Additional tasks to verify the universality of the method]
> We thank the reviewer for the suggestion regarding the generalization of KnowProxy. In response, we have conducted additional experiments on a representative multilingual benchmark, mCSQA [1], to further evaluate the generalizability of KnowProxy in multilingual settings.
> | **Method**               	| **mCSQA (fr)** | **mCSQA (ru)** |
> |------------------------------|:--------------:|:--------------:|
> | Llama 3.2 (3B) (Fine-tuning) |  	72.5      |  	44.6  	|
> | Llama 3.2 (3B) (Zero-shot)   |      64.7  	|  	36.6      |
> | Llama 3.2 (3B) (KnowProxy)   |      69.4  	|  	38.7      |
>
> This addition demonstrates that KnowProxy effectively enhances the multilingual capabilities of LLMs and further supports its ability to generalize beyond the original experimental setup.
>
> We will include these new findings and the corresponding discussion in the revised manuscript to more clearly demonstrate the generalization capability of KnowProxy.
>
> ### [Question1: Utilizing original answer from LLM for KnowProxy]
> We thank the reviewer for the insightful suggestion regarding our framework. Following this suggestion, we conducted experiments comparing (i) the original KnowProxy, where the proxy model utilizes only the elicited knowledge as-is, and (ii) the suggested variant, where the proxy model uses both the LLM-generated knowledge and the LLM’s original answer.
>
> As shown in the table below, KnowProxy with the LLM answer exhibits poorer performance than the proposed method. While this observation requires further analysis, it suggests that incorporating the original answers from LLMs potentially introduce error propagation into the small proxy models, thereby negatively affecting the proxy model’s knowledge adaptation.
> | **Method**       	| **OBQA** | **PIQA** | **StrategyQA** | **SIQA** |
> |----------------------|:--------:|:--------:|:--------------:|:--------:|
> | KnowProxy        	|   85.0   |   87.2   |  	74.7  	|   77.0   |
> | KnowProxy (w/ LLM Answer) |   76.8   |   83.7   |      72.9  	|   76.4   |
>
> We have included these results in the “Ablation study” (Section 4.4). We greatly appreciate the reviewer’s valuable suggestion, which helped strengthen the contribution of each component in KnowProxy.
>
> ### References
> [1] Sakai et al., mCSQA: Multilingual Commonsense Reasoning Dataset with Unified Creation Strategy by Language Models and Humans, Findings of ACL 2024

---

> > ### Author Response · Authors · 2025-11-27
> >
> > Dear Reviewer ZhQu,
> >
> > Thank you for your thoughtful and constructive review on our manuscript.
> >
> > As we prepare our author response, we have made every effort to comprehensively address the concerns raised by the reviewer.
> > To ensure the adequacy of our responses, we kindly request your confirmation or any additional guidance you may have.
> >
> > Best regards,
> >
> > Paper 16389 Authors

---

### Official Review · Reviewer_cLKo · 2025-10-31

**Soundness:** 3
**Presentation:** 3
**Contribution:** 3
**Rating:** 6
**Confidence:** 3

**Summary:**

Direct fine-tuning of LLMs is computationally expensive and impossible for proprietary black-box models. Existing proxy methods require access to LLMs’ probability distributions and suffer from unstable distributions, limiting their applicability. This paper proposes KNOWPROXY, a knowledge-guided proxy framework for adapting Large Language Models (LLMs) without relying on their output probability distributions—addressing key limitations of existing proxy-based LLM adaptation methods.

* KNOWPROXY outperforms existing proxy methods across all benchmarks and matches direct LLM fine-tuning on tasks like OpenBookQA and StrategyQA.
* It works for black-box LLMs (e.g., ChatGPT) and quantized models (e.g., 4-bit Llama-2-13B), with performance scaling with SLM capability.
* Ablation studies confirm filtering and knowledge adaptation are critical (removing adaptation causes the largest performance drop), while dynamic routing reduces inference cost without accuracy loss.
* Cross-LLM transferability: Proxies trained on one LLM (e.g., Llama 3.2) maintain performance when paired with another (e.g., ChatGPT).

**Strengths:**

1.	By replacing LLM probability distributions with textual knowledge, KNOWPROXY enables adaptation for black-box LLMs (only text outputs available) and avoids instability from unreliable distributions—filling a key gap in existing proxy methods.
2.	The dynamic routing mechanism balances accuracy and inference cost by invoking the proxy only when needed, addressing the "always-on proxy" inefficiency of prior work.
3.	Works with diverse LLMs (open-source/black-box/quantized) and SLMs, with cross-LLM proxy transferability (no retraining needed for new LLMs).

**Weaknesses:**

1.	Performance relies heavily on the quality of prompted knowledge-poorly designed prompts or LLM hallucinations (even after filtering) could degrade proxy training. The paper does not explore how prompt variations (beyond task-specific templates) impact results.
2.	The threshold $/alpha$ for knowledge filtering is predefined but not justified (e.g., no sensitivity analysis on how $/alpha$ affects performance across tasks).
3.	The threshold $/tau$ for routing is also predefined, with no discussion of how to optimize $/tau$ for different tasks/LLMs (e.g., whether τ should be task-specific).

**Questions:**

The same as the weakness.

---

> ### Author Response · Authors · 2025-11-22
> **Response**
>
> We thank reviewer cLKo for your thorough review and the valuable comments on our paper.
>
> ### [Weakness1: Performance sensitivity to prompt designs]
> We appreciate the reviewer’s thoughtful observation regarding the potential sensitivity of our framework to the quality of knowledge stemming from prompt design.
>
> To evaluate the impact of prompt styles on KnowProxy, we conducted experiments with diverse prompting strategies. Specifically, we compared our decomposition-style prompting with two representative approaches [1,2]: zero-shot generation and few-shot generation. In the few-shot setting, the LLM is provided with one example from each dataset (i.e., CSQA, QASC, and StrategyQA) to specify the type of knowledge it should generate. In contrast, the zero-shot setting allows the LLM to freely generate knowledge related to the given question without any examples.
> | **Method**            | **CSQA** | **QASC** | **StrategyQA** |
> |-----------------------|:--------:|:--------:|:--------------:|
> | Zero-shot LLM         |   67.7   |   75.9   |      60.9      |
> | KnowProxy (Decomposition) |   75.0   |   78.1   |      72.9      |
> | KnowProxy (Few-shot)  |   74.6   |   78.7   |      66.4      |
> | KnowProxy (Zero-shot) |   74.7   |   77.6   |      72.9      |
>
> These results confirm that, across diverse prompting strategies, prompting with decomposing strategy (used in KnowProxy) consistently achieves highest performance across complex reasoning benchmarks.
>
> We have clarified this point in the revised manuscript and added the impact of prompting engineering in our framework. Thank you for suggesting the important analysis.
>
>
>  ### [Weakness2, 3: Additional analysis on α and τ]
> We appreciate the reviewer’s suggestion to clarify our hyperparameter settings (i.e., the thresholds for knowledge filtering and routing).
>
> Regarding the threshold for the filtering, we conducted a small-scale study across datasets to identify an optimal range for proxy training in the below table. We observe that the optimal \alpha falls within the range of 0.3 to 0.7 for all tasks. Consistent with findings from previous work [3], this result also suggests that excessive knowledge filtering can diminish the effectiveness of knowledge adaptation.
> | **α** | **OBQA** | **PIQA** | **QASC** | **StrategyQA** |
> |:----------:|:--------:|:--------:|:--------:|:--------------:|
> |     0.0    |   76.8   |   79.2   |   74.4   |      70.7      |
> |     0.1    |   74.8   |   80.1   |   73.2   |      65.5      |
> |     0.2    |   76.0   |   80.1   |   73.7   |      67.7      |
> |     0.3    |   76.4   |   80.6   |   74.7   |      70.3      |
> |     0.4    |   76.8   |   79.7   |   75.9   |      67.7      |
> |     0.5    |   77.4   |   80.1   |   75.6   |      69.0      |
> |     0.6    |   77.8   |   79.3   |   73.9   |      70.3      |
> |     0.7    |   73.8   |   79.4   |   75.4   |      73.8      |
> |     0.8    |   74.4   |   80.1   |   74.2   |      66.8      |
> |     0.9    |   72.6   |   79.1   |   71.2   |      66.8      |
> |     1.0    |   64.0   |   74.4   |   62.9   |      62.4      |
>
>
>
> Similarly, the routing thresholds τ are also selected based on empirical observations (in below table). The analysis identifies a potential pattern regarding the routing threshold τ: Llama-series models require relatively low thresholds, whereas ChatGPT necessitates a significantly higher value.
>
> Considering that prior studies [4] characterize Llama as well-calibrated and ChatGPT as prone to overconfidence, this suggests a potential relationship between a model's calibration level and the optimal threshold. It implies that overconfident models may require a stricter (higher) τ to filter out unreliable predictions effectively.
> | **LLM**        | **OBQA** | **ARC-Challenge** | **PIQA** | **CSQA** | **StrategyQA** | **QASC** | **SIQA** |
> |----------------|:--------:|:-----------------:|:--------:|:--------:|:--------------:|:--------:|:--------:|
> | Llama 3.2 (3B) |    1.0   |        0.3        |    1.0   |    0.7   |       0.6      |    0.5   |    1.0   |
> | ChatGPT        |    1.0   |        1.0        |    1.0   |    1.0   |       1.0      |    0.9   |    0.6   |
> | Mistral (7B)   |    1.0   |        0.8        |    1.0   |    0.8   |       1.0      |    1.0   |    1.0   |
>
>
>
>
> We thank the reviewer for this insightful comment, which has strengthened our analysis. We will revise the manuscript to include these empirical findings and the discussion (in Section 3.2, Section 3.3 and Appendix).
>
> ### References
> [1] Liu et al., Generated Knowledge Prompting for Commonsense Reasoning, ACL 2022
>
> [2] Liu et al., Rainier: Reinforced Knowledge Introspector for Commonsense Question Answering, EMNLP 2022
>
> [3] Gekhman et al., Does Fine-Tuning LLMs on New Knowledge Encourage Hallucinations?, EMNLP 2024
>
> [4] Xiong et al., Can LLMs Express Their Uncertainty? An Empirical Evaluation of Confidence Elicitation in LLMs, ICLR 2024

---

> > ### Author Response · Authors · 2025-11-27
> >
> > Dear Reviewer cLKo,
> >
> > Thank you for your thoughtful and constructive comment on our work.
> >
> > As we prepare our author response, we have made every effort to comprehensively address the concerns raised by the reviewer. To ensure the adequacy of our responses, we kindly request your confirmation or any additional guidance you may have.
> >
> > Best regards,
> >
> > Paper 16389 Authors

---

### Author Response · Authors · 2025-12-04
**General Response**

We sincerely thank the reviewers for their thoughtful and constructive reviews.

During the discussion/rebuttal phase, we addressed the reviewers’ specific concerns and suggestions, after which we revised the manuscript as follows:

- **Additional Explanations**: We added more detailed analyses on (i) the impact of prompt designs for eliciting LLM-generated knowledge (Reviewer cLKo and YMsf), (ii) the performance sensitivity to hyperparameters (i.e., α and τ) (Reviewer cLKo and YMsf), and (iii) the justification for aggregating confidence scores (Reviewer YMsf).
- **Extended Results**: We incorporated additional experiments on stronger models to validate the necessity of generating an appropriate amount of knowledge (Reviewer ZhQu), provided ablation studies on utilizing the LLM’s original answers for KnowProxy (Reviewer ZhQu), and included additional experiments on more diverse tasks to demonstrate the generalization capability of KnowProxy (Reviewer ZhQu).
- **Further Discussions**: We expanded our discussion of the overconfidence issue of LLMs (Reviewer YMsf), the transferability of the proxy model (Reviewer YMsf), and the training of KnowProxy (Reviewer Yfto). We also provided more precise and detailed explanations of the effectiveness of KnowProxy (Reviewer ZhQu).

We believe that these revisions comprehensively address the major concerns raised by the reviewers and further strengthen the contribution of our work.

---

### Meta-Review · Area_Chair_zn3c · 2026-01-06

**Summary:**

This paper studies adapting LLMs using a small LLM proxy to new domains without access to the output probability distribution. The authors propose KnowProxy, which trains a lightweight proxy LLM based on the elicited textual knowledge from a frozen large LLM. Overall, all the reviewers agree that this work addresses an important and practical problem, and KnowProxy is a novel and empirically strong LLM adaptation method.

Although some concerns were raised regarding the reliance on knowledge elicitation prompting, sensitivity to the hyperparameters, and generality to updated LLMs and tasks, the authors provide detailed and direct responses to address these concerns. While some limitations remain, such as the reliance on LLM-elicited confidence, mixed results with GPT-5-mini, and evaluated domains, these can be appropriately framed as future directions rather than fundamental flaws. The method offers valuable contributions to the efficient adaptations of LLMs and will be interesting to the community.

**Reviewer Concerns:**

The common concerns raised by reviewers are:
- Hyperparameter sensitivity, specifically concerning the knowledge-elicitation prompting, the robustness of KnowProxy to the choice of the filtering threshold and the routing threshold. The authors provide direct robustness studies to address the concern.
- Evaluation with more powerful LLMs and domains like coding. The authors provide extended results with Qwen3 and GPT-5-mini to demonstrate the effectiveness of the proposed approach. The results partially address the concern. Nevertheless, the extension to other domains does not pose fundamental flaws, given the significant empirical gains on the existing benchmarks.

Overall, the authors did a good job in addressing the main concerns raised by reviewers.

**Reviewer Scores:**

This work initially received a positive evaluation by all reviewers. During the rebuttal, the authors also effectively addressed most of the concerns. Therefore, I am leaning to a positive recommendation of this work.

---

### Decision · Program_Chairs · 2026-01-26

Accept (Poster)